# Activation of mGluR5 and NMDA Receptor Pathways in the Rostral Ventrolateral Medulla as a Central Mechanism for Methamphetamine-Induced Pressor Effect in Rats

**DOI:** 10.3390/biom10010149

**Published:** 2020-01-16

**Authors:** Chih-Chia Lai, Chi Fang, Chung-Yi Kuo, Ya-Wen Wu, Hsun-Hsun Lin

**Affiliations:** 1Department of Pharmacology, School of Medicine, Tzu Chi University, Hualien 970, Taiwan; cclai@mail.tcu.edu.tw; 2Master Program in Medical Physiology, School of Medicine, Tzu Chi University, Hualien 970, Taiwan; betty11708@hotmail.com (C.F.); bibby1987@hotmail.com (C.-Y.K.); ezepzj@yahoo.com.tw (Y.-W.W.); 3Department of Physiology, School of Medicine, Tzu Chi University, Hualien 970, Taiwan

**Keywords:** methamphetamine, pressor effect, rostral ventrolateral medulla, NMDA receptor, mGluR5, PKC

## Abstract

Acute hypertension produced by methamphetamine (MA) is well known, mainly by the enhancement of catecholamine release from sympathetic terminals. However, the central pressor mechanism of the blood-brain-barrier-penetrating molecule remains unclear. We used radio-telemetry and femoral artery cannulation to monitor the mean arterial pressure (MAP) in conscious free-moving and urethane-anesthetized rats, respectively. Expression of Fos protein (Fos) and phosphorylation of N-methyl-D-aspartate receptor subunit GluN1 in the rostral ventrolateral medulla (RVLM) were detected using Western blot analysis. ELISA was carried out for detection of protein kinase C (PKC) activity in the RVLM. MA-induced glutamate release in the RVLM was assayed using in vivo microdialysis and HPLC. Systemic or intracerebroventricular (i.c.v.) administration of MA augments the MAP and increases Fos expression, PKC activity, and phosphorylated GluN1-ser 896 (pGluN1-ser 896) in the RVLM. However, direct microinjection of MA into the RVLM did not change the MAP. Unilateral microinjection of a PKC inhibitor or a metabotropic glutamate receptor 5 (mGluR5) antagonist into the RVLM dose-dependently attenuated the i.c.v. MA-induced increase in MAP and pGluN1-ser 896. Our data suggested that MA may give rise to glutamate release in the RVLM further to the activation of mGluR5-PKC pathways, which would serve as a central mechanism for the MA-induced pressor effect.

## 1. Introduction

Methamphetamine (MA) is a popular and highly addictive psychostimulant that not only affects neurobehavior but also causes cardiovascular dysfunctions including tachycardia, myocardial ischemia, and hypertension [1]. Numerous studies have focused on the psychoactive effects of MA, including arousal, reduced appetite, euphoria, anxiety, and addiction that mainly result from activation of the serotoninergic and dopaminergic system in the CNS [2,3]. It is well known that the MA-induced hypertensive effect is due to the generation of a hyperadrenergic state in the peripheral area [4]. Because of structural similarity, MA acts as a substrate for the noradrenaline transporter and vesicular monoamine transporter 2, leading to an increase in the concentration of the neurotransmitter in the synaptic cleft [1]. Besides, the excessive synaptic level of noradrenaline also results from the inhibition effect of MA on the monoamine oxide [5,6]. A relatively early study showed that MA produced a dose-dependent increase in blood pressure in conscious squirrel monkeys. The α1-adrenoceptor antagonist prazosin completely blocked the pressor effect of MA [7].

Although MA is commonly taken by systemic administration, it can easily penetrate the blood-brain-barrier (BBB), which makes it able to distribute throughout the CNS quickly. MA causes a transient increase in mean arterial pressure (MAP) immediately following intraperitoneal (i.p.) delivery to anesthetized-mice [8]. After the intranasal application of a single-dose of MA to human participants, the peak pressor effect occurred after about 5 min, while the peak plasma level of MA was observed 4 h after MA administration [9]. Rivière et al. [10] reported that the rank order of MA tissue accumulation was kidney > spleen > brain > liver > heart > serum after intravenous (i.v.) injection of MA to anesthetized rats. Interestingly, the same study indicated that the brain/serum concentration ratio rose from 7:1 at 2 min to a peak of 13:1 at 20 min. A similar result, reported by Li et al. [11], showed that the distribution of MA to the brain tissue was significantly higher than to the heart 20 min after i.v. administration. Furthermore, MA shows a preferential distribution to brainstem nuclei that are associated with cardiovascular regulation. These findings inspire us to consider the fact that systemic application of MA may act firstly on the brain presympathetic vasomotor nuclei, leading to its initial hypertension.

The rostral ventrolateral medulla (RVLM), a central sympathoexcitatory nucleus containing many presympathetic neurons, is well known to be critical to the sympathetic regulation of blood pressure (BP) [12]. We hypothesize that the RVLM may serve as a logical neural substrate for the pressor effect of MA. Glutamate is the main neurotransmitter involved in cardiovascular regulation in the RVLM [13]. Activation of N-methyl-D-aspartate receptors (NMDARs) in the RVLM contributes to the elevation of BP via an increase in the sympathetic outflow [14]. Given that phosphorylation of NMDAR subunits increases the receptor function [15], and that GluN1 subunits are the obligatory subunit of the functional NMDARs, we detected the phosphorylated state of GluN1 in the RVLM after MA exposure. Our data demonstrated that MA indeed augmented the glutamate amount and phosphorylated GluN1-ser 896 (pGluN1-ser 896) in the RVLM by intracerebroventricular (i.c.v.) application, even though it did not affect BP when it was introduced into the RVLM directly. The results indicated that MA might evoke glutamate release in the RVLM. Afterwards, the glutamate activates metabotropic glutamate receptor 5 (mGluR5) in the RVLM to initiate the protein kinase C (PKC) signaling cascade and the underlying phosphorylation of NMDAR, leading to pressor responses.

## 2. Materials and Methods

### 2.1. Animals

All animal care and experimental procedures were carried out in accordance with the protocol approved by the Institutional Animal Care and Use Committee of Tzu Chi University (protocol nos. 97017 and 106071). At the Laboratory Animal Center, Tzu Chi University, male Sprague Dawley (SD) rats (BioLASCO Co., LTD., Taipei, Taiwan), weighing 300–350 g, were housed two per cage with food and water available ad libitum under a 12-h/12-h light/dark cycle, with lights on at 7:00 a.m. The animals were allowed to acclimatize for at least 7 days prior to experimental manipulation. Some animals were used for the assessment of cardiovascular function, and some animals were used for neurochemical studies. Each of the neurochemical studies, including Western blot analysis, ELISA, or HPLC detection used an independent group of animals. Efforts were made to minimize suffering and reduce the number of animals used in the experiments.

### 2.2. Measurement of Blood Pressure and Heart Rate

For conscious free-moving rats, we used a radio-telemetry system (Data Sciences International, St. Paul, MN, USA) to record BP and heart rate (HR), as described previously [16]. Under sodium pentobarbital (60 mg/kg, i.p.) anesthesia, the telemetry catheter of a pressure transmitter (model TA11PA-C40) was implanted into the abdominal aorta below the renal artery. The tip of the probe catheter was inserted rostrally through a tiny hole in the abdominal aorta and fixed using sterile cotton tip applicators and a small amount of surgical glue (3M Vetbond Tissue adhesive, Animal Care Products; St. Paul, MN, USA). The transducer unit was positioned into a cavity along the abdomen and was sutured to the inside of the muscle wall. After surgery, each rat received antibiotic chlortetracycline for the wound and a nonsteroidal anti-inflammatory drug (carprofen, 5 mg/kg, subcutaneous injection, per 24 h for 7 days). Animals were housed in individual cages for 7 days for recovery. At the time of the BP and HR measurements, each cage, one rat in each, was placed over the receiver panel connected to the computer for continuous data acquisition over 130 min, which was averaged every 10 min. The expert researcher gently grasped the rat and exposed its abdomen. The i.p. injection was performed carefully to avoid affecting the telemetry probe. Each rat received only a single injection of saline or MA. The injection volume was 1 mL/kg. Each i.p. injection was accomplished within 10 s. In the set of experiments, BP and HR were recorded before and after i.p. injection of saline or MA.

For anesthetized groups, rats were injected with urethane (1.2 g/kg, i.p.) first, and additional urethane (0.3 g/kg, i.p.) was applied if the rats responded to tail pinch. The procedures for recording BP and HR were similar to our recent report [17]. The left femoral artery was cannulated with a polyethylene tubing (PE 50) and connected to a pressure transducer with its output to a Gould EasyGraf Recorder (TA420) (Valley View, OH, USA) for the recording of BP and HR. The signals from the recorder were sent to a data acquisition system (MP 100, BIOPAC System, Inc., Santa Barbara, CA, USA) for continuous recording, and the built-in function of the acquisition system provided simultaneous measurements of MAP. In the set of experiments, BP and HR were recorded before or after intracerebroventricular (i.c.v.) or intra-RVLM injection of MA.

### 2.3. Intracerebroventricular Cannula Implantation

Under sodium pentobarbital (45 mg/kg, i.p.) anesthesia, rats were placed prone in a David Kopf (Tujunga, CA, USA) stereotaxic frame with an incisor bar, and a parietal hole was drilled to the desired position in relation to the bregma, as described below. A stainless steel guide cannula (23-gauge) was stereotaxically placed into the right lateral cerebroventricle using the following stereotaxic coordinates: 1.6 mm lateral to the midline, 0.8 mm caudal to the bregma, and 3.5 mm below the dorsal surface of the brain. The cannula was secured to the skull using two stainless steel screws and dental cement and closed with a removable stylet. Animals were allowed a five-day recovery period prior to the experiment. Carprofen (5 mg/kg, subcutaneous injection, per 24 h for 3 days) was applied to avoid pain caused by inflammation. For i.c.v. injections, a 27-gauge stainless steel micropipette was introduced into the ventricle through the cannula and connected to Hamilton microsyringes (10 μL); drugs (5 μL) were injected using a syringe pump (KDS 100) at a rate of 1.6 μL/min. To verify the correct placement of the guide cannula, the drinking behavior elicited by i.c.v. angiotensin II (50 ng, 5 μL) 3 days after surgery was observed; the rats showing drinking behavior within 1 min after the injection were used for the experiments. Only two rats failed to pass the angiotensin II test in this study. For detecting the pressor effect of i.c.v. injection of MA, a set of animals was injected with both saline and MA. MA (50, 150, or 500 nmol) was injected 30 min after saline injection. For assessing whether the repeated i.c.v. injection of MA did induce a reproducible pressor effect, one set of animals was injected with MA (150 nmol) four times at intervals of 30 min. The other two sets of animals were used to evaluate the effects of microinjections of kinase inhibitor or mGluR5 antagonist into the RVLM on the pressor effect of i.c.v. injection of MA.

### 2.4. Microinjection into the RVLM

The procedures for the RVLM microinjections and microdialysis were similar to our previous study [18]. Five days after i.c.v. cannula implantation, unilateral microinjection into the RVLM was executed under urethane anesthesia. After a rat had finished the set-up of BP recording, it was arranged on the stereotaxic frame as it had been for implanting the i.c.v. cannula. An interparietal hole was drilled to the desired position in relation to the lambda, as described below. A 30-gauge stainless steel micropipette was introduced into the RVLM through the hole and connected to Hamilton microsyringes (0.5 μL) for injection. Coordinates for the micropipette were 1.9–2.0 mm left lateral to the midline, 3.0 mm caudal to the lambda, and 7.5–8.0 mm below the dorsal surface of the brain [19]. All microinjections were made at a volume of 0.1 μL to the unilateral RVLM and were finished within 10 s. NMDA (0.14 nmol) was administered into the injection side to identify the RVLM functionally at the end of each BP-recording experiment. The requirement for data collected in this study was that NMDA induced a pressor effect bigger than 10 mmHg when it was applied into the RVLM at the end of each experiment. In the present study, all rats satisfy the criterion.

### 2.5. Microdialysis and HPLC for Detect Glutamate Concentration in the RVLM

For the microdialysis experiment, the dialysis probe (A-I-12-01, Eicom Corp., Tokyo, Japan) was inserted into the RVLM region using the same coordinates as the microinjection. The experiment was performed by perfusing the probe continuously with artificial cerebrospinal fluid (aCSF) at a rate of 1 μL/min with a syringe pump (KDS 100). The aCSF consists of (in mM) 148.0 NaCl, 3.0 KCl, 1.4 CaCl_2_·2H_2_O, 0.8 MgCl_2_, 0.8 Na_2_HPO_4_, and 0.2 NaH_2_PO_4_·H_2_O. The dialysate was collected at 10 min intervals, and the total volume of each dialysate sample was 10 μL. The glutamate level in the sample was immediately assayed with HPLC-ECD (HTEC-500, Eicom Corp., Japan) using L-glutamate as the standard. When glutamate levels were stable and consistent for two consecutive measurements, the experiments with MA injection then proceeded. The detail protocol for i.c.v. accompanying microdialysis from the RVLM is shown in Figure 7.

### 2.6. Western Blotting for Detecting Fos protein (Fos) Expression and Phosphorylated GluN1

The procedure for Western blot analysis of brain tissue was similar to that described in earlier studies [20]. Rats were sacrificed, and brains were rapidly removed and soaked in ice-cold Krebs solution for 1 min. The brainstems were isolated from the brain and quickly frozen by cold spray (FREEZE 75; CRC Industry Europe NV, Zele, Belgium). A coronal section, 0.5–1.5 mm rostral to the obex, was prepared from the brainstem. Both sides of the ventrolateral regions (about 2 mm to the midline) of the slice from each rat were punched out by a tissue puncher (1 mm in diameter), and the small punched-out tissue was identified as the RVLM. The medulla (CVLM) was taken out in the same way as the RVLM, except that the coronal section was made 0.5 mm rostral and caudal to the obex. The isolated tissues were frozen in liquid nitrogen and stored at −85 °C until use. The tissues were homogenized in 30 μL lysis buffer consisting of 0.32 M sucrose, 1 mM EDTA, and 1 mTIU/mL aprotinin using a homogenizer at a speed of 10,000 rpm for 2 × 30 s. SDS was added to the sample to a final concentration of 0.1%, and 20 μg of protein was electrophoresed on 8% denaturing polyacrylamide gels. Separated proteins were transferred to nitrocellulose membrane and probed with the primary antibody. The bound antibody was incubated with secondary goat anti-rabbit antibody (1:3000, Santa Cruz Biotechnology, Santa Cruz, CA, USA) conjugated to horseradish peroxidase, which was measured with Western Blotting Luminol Reagent (Santa Cruz Biotechnology, Santa Cruz, USA). The chemiluminescent signal was detected by X-ray film (Fuji Photo Film Co., Ltd., Tokyo, Japan), and the intensity of the bands was digitalized by scanner and analyzed with UN-SCAN-IT gel software version 6.1 for Windows (Silk Scientific Corporation, Orem, UT, USA). Protein concentrations were determined by the bicinchoninic acid method (Sigma Co., St. Louis, MO, USA) using bovine albumin as the standard. To detect the phosphorylation of GluN1, anti-GluN1 (1:3000), anti-phopho-GluN1-ser 896 (1:3000), and anti-phopho-GluN1-ser 897 (1:3000), purchased from Upstate (Lake Placid, NY, USA), were used as primary antibodies. Rats were sacrificed 30 min after saline or MA were administered intraperitoneally. In the groups of i.c.v. application of MA or saline, rats were decapitated 5 min after the drugs were applied. In the assay of Fos expression, the primary antibodies selected were anti-Fos (1:500, Santa Cruz Biotechnology, Santa Cruz, USA) and anti-β-actin (1:10,000, Millipore, Temecula, CA, USA). In this set of tests, animals were sacrificed 2 h after MA or saline were applied.

### 2.7. Determination of PKC Activity

The relative PKC activity of the RVLM was measured using an ELISA kit (Enzo Life Sciences International, Inc., Farmingdale, NY, USA), following the manufacturer’s protocol. In the assay, microtiter plates pre-coated with PKC substrate were used. The plate wells were soaked with kinase assay dilution buffer and were emptied after 10 min at room temperature. An equal volume of the tissue supernatants was added to the wells, followed by the addition of ATP to initiate the reaction and incubation for 90 min at 30 °C in a water bath. The kinase reaction was terminated by pouring out the contents of each well, and then a phospho-specific substrate antibody, as the primary antibody, which binds specifically to the phosphorylated peptide substrate, was added to the wells. The phospho-specific antibody was subsequently bound by a peroxidase-conjugated secondary antibody. After that, tetramethylbenzidine substrate was added to each well and incubated at room temperature for 30–60 min depending on the color development. Finally, using an acid stop solution to stop the color development, the intensity of the color was measured in a microplate reader at 450 nm. For the i.p. injection MA group, rats were sacrificed 30 min following saline or MA administration, and the RVLM regions were carefully punched out from the brain slice for Western blot experiments. For the i.c.v. injection MA group, rats were sacrificed 5 min after MA was applied, and the RVLM regions were taken out immediately.

### 2.8. Data Analysis

Data were expressed as mean ±SEM and were plotted and analyzed statistically with GraphPad Prism version 6.0 for Windows (GraphPad Software, San Diego, CA, USA). The time course of changes in MAP or HR by different doses of MA or saline was analyzed using two-way ANOVA followed by the Bonferroni’s multiple comparison post-test for comparing the corresponding time points of the saline-treated group. The results of Western blots and PKC activity studies were analyzed by unpaired *t*-test (comparison of two groups) or one-way ANOVA followed by the Bonferroni’s multiple comparison post-test (comparison of three groups). The results of i.c.v. saline following an MA-induced increase in blood pressure were analyzed using a paired *t*-test, but the pressor effect induced by the different doses of MA were analyzed by one-way ANOVA followed by Bonferroni’s multiple comparison post-test. The effect of Bisindolylmaleimide II (BIM) and 2-methyl-6-(phenylethynyl) pyridine hydrochloride (MPEP) on BP and glutamate concentration assay were analyzed using one-way ANOVA repeated measures followed by Bonferroni’s multiple comparison post-test.

### 2.9. Chemical Agents

MA was purchased from the National Bureau of Controlled Drugs, Department of Health, Taipei, Taiwan. Pentobarbital sodium came from SCI-Pharmtech (Taoyuan, Taiwan). The PKC kit was bought from Enzo Life Sciences, Inc (Farmingdale, NY, USA). Bisindolylmaleimide II (BIM) and 2-methyl-6-(phenylethynyl) pyridine hydrochloride (MPEP) were obtained from Tocris Bioscience (Bristol, UK). NMDA and other chemicals were purchased from Sigma Co. (St. Louis, MO, USA). Aprotinin and the other reagents used for Western blot analysis were also from Sigma Co. Reagents for electrophoresis were obtained from Bio-Rad Laboratories (Richmond, CA, USA). Stock solutions of BIM and MPEP were prepared in dimethyl sulfoxide (DMSO) from J. T. Baker (Phillipsburg, NJ, USA), and were then diluted to given a concentration with saline. The other agents were dissolved in saline.

## 3. Results

### 3.1. i.p. MA-Induced a Pressor Effect in Conscious Free-Moving Rats

i.p. injection of MA (2 and 10 mg/kg) induced a significant increase in MAP in a dose-dependent manner in conscious free-moving rats (Figure 1A). The pressor effect occurred immediately following the injection in both groups and lasted for a 120-min observation period in the high dose group. MA did not produce a noticeable change in HR until 90 min after the injection (Figure 1B). Nineteen rats were used in the set of tests. The resting mean MAP was 98.86 ± 1.37 mmHg, and the mean HR was 412.0 ± 8.73 bpm. In the MAP analysis groups, two-way ANOVA showed a significant treatment effect with F(2,16) = 16.70 and *p* = 0.0001, time effect with F(13,208) = 5.070 and *p* < 0.0001, and interaction between treatment and time with F(26,208) = 9.628 and *p* < 0.0001. In the HR analysis groups, two-way ANOVA showed the treatment effect with F(2,16) = 0.8792 and *p* = 0.4342, time effect with F(13,208) = 1.479 and *p* = 0.1270, and interaction between treatment and time with F(26,208) = 6.6563 and *p* < 0.0001.

### 3.2. i.p. MA Increased FOS Expression and Phosphorylation of GluN1-ser 896 in the RVLM in Conscious Free-Moving Rats

Even though FOS expression takes time, it is an important marker to backtrack whether or not the neurons were activated. After 2 h i.p. injection of MA, the FOS/β-actin ratio significantly and specifically increased in the RVLM but not in the CVLM (Figure 2A,B). One-way ANOVA showed the treatment effect with F(2,9) = 24.42 and *p* = 0.0002 in the RVLM group. The mean FOS/β-actin ratio in the RVLM was 100.0 ± 11.56 in saline (*n* = 4), 163.69 ± 5.24 in MA 2 mg/kg (*n* = 4), and 212.53 ± 15.17 in MA 10 mg/kg (*n* = 4). The ratio of pGluN1-ser 896 to GluN1 was also significantly higher after i.p. injection of 2 or 10 mg/kg MA (255.48 ± 29.64 or 264.97 ± 35.38, *n* = 6 in each group) than the injection of saline (100.0 ± 4.88, *n* = 6) in the RVLM (Figure 2C). One-way ANOVA showed the treatment effect with F(2,15) = 11.95 and *p* = 0.0008 in the ser-896 group. However, the ratio of pGluN1-ser 897 to GluN1 in the RVLM had no significant change after MA was injected intraperitoneally (Figure 2D).

### 3.3. i.c.v. MA-Induced a Pressor Effect in Urethane-Anesthetized Rats

For mimicking the central distribution of systemic administration of MA, we applied MA by i.c.v. injection in urethane-anesthetized rats. MA caused a significantly dose-dependent pressor effect (Figure 3A,B). The BP was raised with a quick onset and reached a peak within 2 min, and the pressor response endured for 10–15 min. The mean of the peak value of the increase in MAP was 5.67 ± 1.15, 10.67 ± 1.23, and 12.67 ± 0.72 mmHg for MA 50, 150, and 500 nmol (*n* = 6 in each group), respectively. One-way ANOVA showed a significant dose effect with F(2,15) = 11.70 and *p* = 0.0009. The original MAP was 79.33 ± 3.42 mmHg (*n* = 18). The magnitude of increase in MAP was reproducible when MA (150 nmol) was applied intracerebroventricularly four times at intervals of 30 min. The resting MAP was 79.44 ± 1.98 mmHg in this set of animals (Appendix A).

### 3.4. i.c.v. MA Increased FOS Expression and Phosphorylation of GluN1-ser 896 in the RVLM in Urethane-Anesthetized Rats

One-way ANOVA showed a significant dose effect with F(2,21) = 4.254 and *p* = 0.0281 in the Fos/β-actin ratio in the RVLM after i.c.v. application of MA in anesthetized rats (Figure 4A). The mean Fos/β-actin ratio was 100.0 ± 4.17, 110.70 ± 7.06, and 143.30 ± 17.05 for saline, MA 50 nmol, and MA 150 nmol treatment, respectively (*n* = 8 in each group). Injection of MA 150 nmol significantly augmented the expression of pGluN1-ser 896 (Figure 4B). The ratio of pGluN1-ser 896 to GluN1 was 100.0 ± 18.92 for saline (*n* = 6) and 203.20 ± 30.52 for MA (*n* = 6).

### 3.5. MA Increased PKC Activity in the RVLM

We recently found that the PKC signaling pathway is implicated in the blood pressure regulation by an increase in the expression of phosphorylated GluN1 in the sympathetic preganglionic neurons (SPN) [21]. Since NMDAR activity is regulated by its phosphorylation state [22], and activation of NMDAR may modulate the neuronal activity in the RVLM [13], we evaluated the PKC activity in the RVLM after i.p. and i.c.v. application of MA in conscious and anesthetized rats, respectively. Our data showed that PKC activity was significantly higher in MA-treated animals compared with the saline-treated groups. In the i.p. group, the relative activity of PKC was 100.0 ± 5.84 (for saline, *n* = 5) and 152.5 ± 3.76 (for MA, *n* = 5) (Figure 5A). In the i.c.v. group, PKC activity was 100.0 ± 2.06 (for saline, *n* = 5) and 138.40 ± 10.87 (for MA, *n* = 5) (Figure 5B).

### 3.6. The PKC Inhibitor Attenuated MA-Induced Increase in MAP and pGluN1-ser 896

In order to further confirm the PKC signaling pathway on the effect of MA (150 nmol), a selective PKC inhibitor BIM was injected unilaterally into the RVLM in anesthetized rats. After 10 min pretreatment of BIM with a dose of 4 nmol, the MA-induced pressor effect was attenuated significantly. Thirty minutes later, the pressor effect of MA recovered. Repeated measures of one-way ANOVA showed a significant time effect with F(2,10) = 9.106 and *p* = 0.0056. Administration of a lower dose of BIM (40 pmol) did not affect the MA-induced increase in MAP with F(2,10) = 1.566 and *p* = 0.2561 (Figure 6A). The amplitude of increase in the MAP of MA were 11.0 ± 0.86, 9.83 ± 1.72, and 9.17 ± 1.33 for 20 min before, 10 min after, and 40 min after BIM (40 pmol, *n* = 6), respectively. The values were 12.17 ± 0.98, 6.0 ± 1.00, and 10.17 ± 1.80 in the BIM 4 nmol group (*n* = 6). The original MAP was 75.83 ± 2.58 mmHg (*n* = 12). Ten minutes before i.c.v. injection of MA, we applied BIM (4 nmol) into the unilateral RVLM in six rats tested. The RVLM tissue was collected 5 min after MA, and it showed that the BIM injected side existed significant lower pGluN1-ser 896 than the non-BIM side. The ratio of pGluN1-ser 896 to GluN1 was 100.0 ± 2.59 for the non-BIM side and 52.83 ± 5.26 for the BIM side. However, BIM could not affect the ratio of pGluN1-ser 896 to GluN1 in the RVLM if rats (*n* = 3) were injected intracerebroventricularly with saline instead of MA. The ratio was 100.0 ± 1.22 and 94.93 ± 4.47 for the non-BIM side and the BIM side, respectively (Figure 6B). In another set of experiments, NMDA (0.14 nmol) was unilaterally microinjected into the RVLM 30 min before and 30 and 60 min after ipsilateral intra-RVLM application of BIM (4 nmol). BIM did not affect the pressor effect induced by NMDA (*n* = 3).

### 3.7. i.c.v. Administration of MA increased the Glutamate Concentration in the RVLM

Even though systemic or i.c.v. administration of MA indeed caused the increase in Fos expression and the ratio of pGluN1-ser 896 to GluN1 in the RVLM, surprisingly, direct microinjection of MA (2 or 20 nmol, *n* = 6 for each group) into the RVLM did not cause any noticeable changes in MAP. In view of glutamate being considered the most important excitatory neurotransmitter in the RVLM, we hypothesized that i.c.v. injection of MA might activate some other areas in the CNS, leading to an increase in glutamate release in the RVLM. Therefore, we measured glutamate concentration before and after i.c.v. injection of MA in the RVLM. The result showed that glutamate concentration significantly increased in the first 10-min period after i.c.v. injection of MA (150 nmol) (Figure 7). Repeated measures of one-way ANOVA shows a significant time effect, with F(3,12) = 36.39 and *p* < 0.0001. The glutamate concentration in the RVLM 10 min-period before, and in every 10-min period after, i.c.v. injection of MA were 0.141 ± 0.043, 0.431 ± 0.070, 0.202 ± 0.067, and 0.223 ± 0.071 μM (*n* = 5).

### 3.8. The mGluR5 Antagonist Attenuated MA-Induced Increase in MAP and pGluN1-ser 896

Because it has been known that the activation of postsynaptic mGluR5 will enhance the NMDAR-mediated response via PKC, mGluR5 is responsive to the stimulation of glutamate [23]. Once we made sure that MA did indeed increase the glutamate concentration in the RVLM, we tried to figure out whether the MA-induced increase in the pGluN1-ser 896 and MAP would be attenuated if we blocked the mGluR5 by unilateral microinjection of a selective mGluR5 antagonist, MPEP, into the RVLM. Our results showed that the relative PKC activity in the RVLM after i.c.v. injection of 150 nmol MA was significantly lower in the MPEP (1 nmol) pretreated side than in the non-MPEP pretreated side. The value for the former was 39.20 ± 5.37, and the latter was 100.0 ± 7.91 for the three rats tested (Figure 8A). MPEP (1 nmol) did not affect the ratio of pGluN1-ser 896 to GluN1 in the RVLM after i.c.v. application of saline (*n* = 3). However, MPEP significantly reduced the i.c.v. injection of MA-induced increase in the expression of pGluN1-ser 896 (Figure 8B). The ratio of the MPEP pretreated side was 74.86 ± 6.17, and the non-MPEP pretreated side was 100.0 ± 2.13 for the three rats tested. The results also showed that MPEP dose-dependently attenuated the i.c.v. injection of the MA-induced pressor effect (Figure 8C). Two-way ANOVA showed a significant dose effect with F(1,7) = 7.399 and *p* = 0.0298. MPEP with a dose of 0.1 nmol decreased the MA-induced increase in the MAP from 11.5 ± 0.5 to 8.75 ± 1.11 mmHg and recovered to 10.50 ± 1.26 mmHg 40 min after MPEP pretreatment (*n* = 4). A high dose of MAEP (1 nmol) reduced the pressor effect of MA from 10.4 ± 0.4 to 5.2 ± 0.2 mmHg, and the pressor effect partially recovered to 8.2 ± 0.2 mmHg 40 min after MPEP pretreatment (*n* = 5). Repeated measures of one-way ANOVA showed a significant time effect with F(2,6) = 10.33 and *p* = 0.0114 in the 0.1 nmol MPEP group, and F(2,8) = 85.17 and *p* < 0.0001 in the 1 nmol MPEP group. The original MAP was 89.67 ± 2.72 (*n* = 9) in this set of animals. In another set of experiments, NMDA (0.14 nmol) was unilaterally microinjected into the RVLM 30 min before and 30 and 60 min after ipsilateral intra-RVLM application of MPEP (1 nmol). MPEP did not produce significant changes in the NMDA-induced increase in MAP (*n* = 3).

## 4. Discussion

The present study showed, for the first time, the causal link between the phosphorylation of GluN1 serine-896 residue in the RVLM and the MA-induced acute pressor effect. The findings provide a novel understanding of the central mechanism of MA that may give rise to glutamate release in the RVLM and cause the activation of the mGluR5-PKC signaling pathways, resulting in a pressor effect.

We demonstrated that, no matter the systemic or central application, MA induced an increase in the MAP immediately following administration. The results are consistent with other studies carried out on conscious animals [7,24,25] and on anesthetized animals [8,26]. On the other hand, our results are not in agreement with some other studies, which showed that MA produced a depressor effect [11,27]. Since the acute clinical effect of MA was found to increase blood pressure [1], the same pressor performance in the systemic and central administration of MA in the current study is of benefit in order to study the central hypertensive mechanism of MA.

Even though a large number of experimental studies have been established to investigate the neuropharmacological effects of MA, there has still been no attempt made to explore MA’s effect by central application directly. In the present study, we conducted the i.c.v. injection of MA to evaluate MA’s central cardiovascular impact separately from its peripheral effects. Therefore, the most important thing at this time is to determine the reasonable dose used in the i.c.v. application of MA. A meaningful study demonstrated that MA dose-dependently increased the MAP, with peaks occurring approximately 5–15 min after MA administration, while the plasma levels were still quite low [9]. An earlier study showed that brain-to-serum ratio rose from 7:1 at 2 min to a peak of 13:1 at 20 min in rats that received i.v. bolus (1 mg/kg) of MA [10]. According to these studies, we formulated the i.c.v. applied dose of MA, which would make the concentration of MA in the brain ten times more than in the serum. Melega et al. [28] assayed the blood samples obtained from 105 criminal individuals, then estimated the initial MA body burdens by pharmacokinetic calculations. Their data showed that the plasma concentrations of MA were in the range of 1 to 20 μM, corresponding with low to high MA-users; the brain MA level was determined at the interval from 10 to 200 μM in the study. The mechanisms of MA-induced increases in BBB permeability, and the disruption of the BBB structure, have been fully discussed [29]. Given that MA is a highly lipophilic molecule [30], we postulate that, while being injected intracerebroventricularly, it will enter the parenchyma rapidly, firstly by crossing the blood-CSF barrier formed by the choroid plexus epithelium and then by penetrating the BBB, which is comprised of microvascular endothelium [31]. With MRI study, the whole rat brain volume (excluding the cerebellum and olfactory bulb) is estimated to be about 1.6 mL [32]. Therefore, the brain concentration of MA was near 94 μM, corresponding to our design concentration while we applied 150 nmol MA in the lateral cerebroventricle.

As mentioned above, MA enters the brain rapidly, and its pressor response is related to an increase in sympathetic activity. Therefore, we selected the RVLM to be our main target. Fos expression increased specifically and significantly in the RVLM, confirming our expectation that the RVLM neurons are activated following MA treatment. Since glutamate is the primary excitatory transmitter in the RVLM and its NMDAR participates in the excitatory response, we used the phosphorylated status of NMDAR as the index of activation of the RVLM neurons [13]. Functional NMDARs are commonly composed of two GluN1 subunits and two identical or different GluN2 (usually 2A or 2B) subunits [33]. Phosphorylation of NMDAR subunits will increase the NMDAR activity in general. In the obligatory subunit, GluN1, serine-890, and serine-896 residues are phosphorylated by PKC, while protein kinase A phosphorylates a neighboring site, serine-897 [22]. As expected, the pressor effect of MA was accompanied by specific augmentation of pGluN1-ser 896 and raising PKC activity in the RVLM. Antagonized PKC in the RVLM decreased the level of pGluN1-ser 896 and attenuated i.c.v. MA-induced pressor response. Considering the fact that the dosage of microinjection will quickly spread from the original injection site, we utilized the application of a relatively high dose of MA into the RVLM. However, the intra-RVLM application of MA did not provoke any marked pressor effect. This clearly indicated that MA might act elsewhere in the CNS, and the RVLM effects are downstream of the direct target. Certainly, glutamate might act on both AMPA and NMDA ionotropic receptors in the RVLM. In consideration of the activation of NMDA receptors, which would induce calcium influx, and the subsequent changes in signaling pathways, MA’s impact on the NMDA receptor may have ongoing effects on neuronal function even though no changes in the BP are observed.

More and more research has found that MA augments glutamate concentrations in the CNS in either acute or chronic studies. An in vivo microdialysis experiment showed that i.p. MA produced a gradual and significant increase in extracellular glutamate concentration in the lateral striatum. VGluT1 might participate in the glutamate release after MA is applied [34]. In human MA-dependent users, the glutamate levels in the brainstem are significantly elevated, and the levels are associated with the duration and dose of MA use [35]. Using the whole-cell patch-clamp recording in rat primary cultured hippocampal neurons, the authors found that MA modulated presynaptic glutamate release in a dopamine-independent manner [36]. Here, we demonstrated that the glutamate concentration was increased in the RVLM, with the corresponding time course of pressor effect induced by i.c.v. administration of MA. Not only ionotropic receptors, but also glutamate, activates the mGluR, which are divided into three groups. Activation Group I receptor will increase the activity of phospholipase C, which leads to the activation of PKC [37,38], while activation of Groups II and III will inhibit adenylyl cyclase [39]. These receptors are all present in the RVLM [40]. Group I mGluR consists of two subunits, mGluR1 and mGluR5. Numerous studies indicate that the activity of mGluR5 is related to addiction to drugs [41]. In the present study, intra-RVLM injection of the mGluR5 antagonist not only decreased the PKC activity and the level of pGluN1-ser 896 in the RVLM but also declined the amplitude of pressor response evoked by i.c.v. MA.

Future works are required to uncover the nuclei releasing the glutamate in the RVLM. The paraventricular nucleus of hypothalamus (PVN) has been recognized as one of the major origins of glutamatergic innervation to the RVLM [42,43], which lead us to reason that the PVN is a logical neural substrate for MA action. In addition, we do not rule out other CNS nuclei that will serve as MA’s targets because MA may distribute all over the brain while it is applied either systemically or intracerebroventricularly. Moreover, we did focus on the glutamate in the RVLM for the MA’s effect in the present study. However, dopamine is also an important object that cannot be ignored. We especially emphasized the role of PKC in the MA-induced hypertensive effect while the PKC has been found to mediate phosphorylation of the dopamine transporter and subsequently regulate synaptic dopamine release [44]. Furthermore, ionotropic glutamate receptors and mGluRs are close to each other in the postsynaptic membrane [45] and may crosstalk to each other to shape their output information [46]. In summary, studies are required to clarify the source of glutamate and the character of dopamine in the central mechanism of the MA-induced pressor effect, and the interaction between ionotropic glutamate receptors and mGluRs will also be an interesting issue for further study.

## 5. Conclusions

In this study, we demonstrated a unidirectional signal that MA facilitates the release of glutamate in the RVLM. Glutamate activates the mGluR5 and then causes an increase in the activity of the PKC in the RVLM. PKC phosphorylates NMDAR subunit GluN1. Finally, the signal cascade results in the hypertensive response.

## Figures and Tables

**Figure 1 biomolecules-10-00149-f001:**
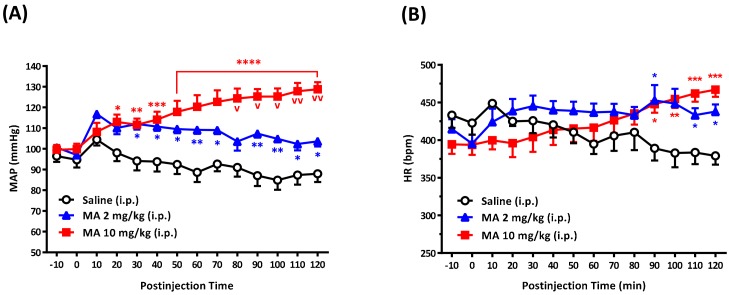
Effects of methamphetamine (MA) on mean arterial pressure (MAP), heart rate (HR), in conscious Sprague Dawley rats. Time-course in MAP (**A**) and HR (**B**) in rats 10 min before and 120 min after intraperitoneal (i.p.) injection of saline (*n* = 6), MA 2 mg/kg (*n* = 5), and MA 10 mg/kg (*n* = 8). Values are mean ± SEM, * *p* < 0.05, ** *p* < 0.01, *** *p* < 0.001, **** *p* < 0.0001 versus the corresponding time of the saline group or ^v^
*p* < 0.01, ^vv^
*p* < 0.0001 versus the corresponding time of the MA 2 mg group analyzed by two-way ANOVA measures followed by Bonferroni’s multiple comparison post-test.

**Figure 2 biomolecules-10-00149-f002:**
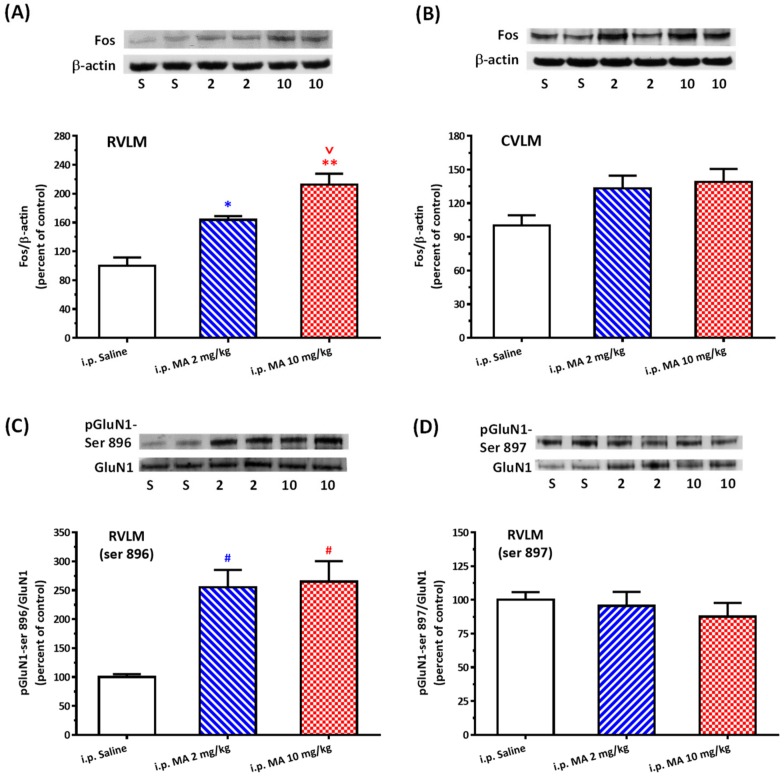
Effects of intraperitoneal (i.p.) injection of methamphetamine (MA) on Fos expression and phosphorylation of GluN1 serine residue in conscious Sprague Dawley rats. (**A**) Top panel shows representative Western blot analysis of the levels of Fos and β-actin in the rostral ventrolateral medulla (RVLM) 2 h after i.p. injection of saline (S), MA 2 mg/kg (2), and MA 10 mg/kg (10). The percentage changes in the ratio of FOS to β-actin are shown in the bottom graph. The ratio treatment with saline is taken as a control of 100%. Values represent the mean +SEM, n = 4 animals in each group. * *p* < 0.05, ** *p* < 0.001 versus saline, and ^v^
*p* < 0.05 versus MA 2 mg/kg analyzed by one-way ANOVA followed by Bonferroni’s multiple comparison post-test. (**B**) Similar to the legend of (**A**) except that the tissue was taken from the caudal ventrolateral medulla (CVLM), *n* = 6 animals in each group. (**C**) Top panel shows representative Western blot analysis of the levels of phosphoserine 896 on the GluN1 subunit (pGluN1-Ser 896) and GluN1subunit (GluN1) in the RVLM 30 min after i.p. injection of saline (S), MA 2 mg/kg (2), and MA 10 mg/kg (10). The percentage changes in the ratio of pGluN1-Ser 896 to GluN1 are shown in the bottom graph. The ratio treatment with saline is taken as a control of 100%. Values represent the mean + SEM, *n* = 6 animals in each group. ^#^
*p* < 0.01 versus saline analyzed by one-way ANOVA followed by Bonferroni’s multiple comparison post-test. (**D**) Similar to the legend of (**C**) except that the phosphorylated site is the GluN1 serine 897 residue (pGluN1-Ser 897), *n* = 6 animals in each group.

**Figure 3 biomolecules-10-00149-f003:**
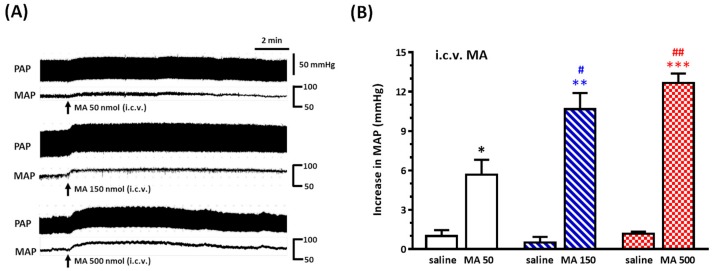
Effects of intracerebroventricular (i.c.v.) injection of methamphetamine (MA) on pulsatile arterial pressure (PAP) and mean arterial pressure (MAP) in urethane-anesthetized Sprague Dawley (SD) rats. (**A**) Representative recordings of i.c.v. MA (50, 150, and 500 nmol) induced an increase in PAP and MAP in three SD rats. (**B**) Bar graph shows the increase in MAP by i.c.v. injection of saline and MA. MA was injected 30 min after saline injection. Values represent the mean + SEM, *n* = 6 animals in each group. * *p* = 0.0306, ** *p* = 0.0005, and *** *p* < 0.0001 versus corresponding saline analyzed by paired *t*-test. # *p* < 0.05 and ## *p* < 0.001 versus MA 50 analyzed by one-way ANOVA followed by Bonferroni’s multiple comparison post-test.

**Figure 4 biomolecules-10-00149-f004:**
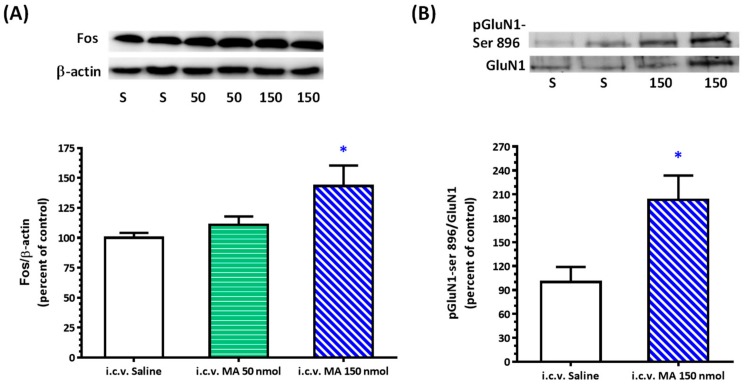
Effects of intracerebroventricular (i.c.v.) injection of methamphetamine (MA) on Fos expression and phosphorylation of GluN1 serine 896 residue in the rostral ventrolateral medulla (RVLM) in urethane-anesthetized Sprague Dawley rats. (**A**) Top panel shows representative Western blot analysis of the levels of FOS and β-actin in the RVLM 2 h after i.c.v. injection of saline (S), MA 50 nmol (50), and MA 150 nmol (150). The percentage changes in the ratio of FOS to β-actin are shown in the bottom graph. The ratio treatment with saline is taken as a control of 100%. Values represent the mean + SEM, *n* = 8 animals in each group. * *p* < 0.05 versus saline analyzed by one-way ANOVA followed by Bonferroni’s multiple comparison post-test. (**B**) Top panel shows representative Western blot analysis of the levels of phosphoserine 896 on the GluN1 subunit (pGluN1-Ser 896) and GluN1subunit (GluN1) in the RVLM 5 min after i.c.v. injection of saline (S) and MA 150 nmol (150). The percentage changes in the ratio of pGluN1-Ser 896 to GluN1 are shown in the bottom graph. The ratio treatment with saline is taken as a control of 100%. Values represent the mean + SEM, *n* = 6 animals in each group. * *p* = 0.0166 versus saline analyzed by unpaired *t*-test.

**Figure 5 biomolecules-10-00149-f005:**
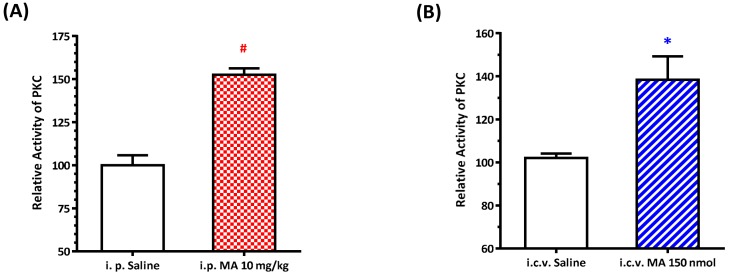
Effects of methamphetamine (MA) on PKC activity in the rostral ventrolateral medulla (RVLM) in Sprague Dawley rats. Bar graphs show the relative activity of PKC in the RVLM 30 min after intraperitoneal (i.p.) injection of saline or MA 10 mg/kg in conscious rats (**A**), and 5 min after intracerebroventricular (i.c.v.) injection of saline or MA 150 nmol in urethane-anesthetized rats (**B**). Values represent the mean + SEM, *n* = 5 animals in each group. * *p* = 0.0111 and # *p* < 0.0001 versus the corresponding saline analyzed by unpaired *t*-test.

**Figure 6 biomolecules-10-00149-f006:**
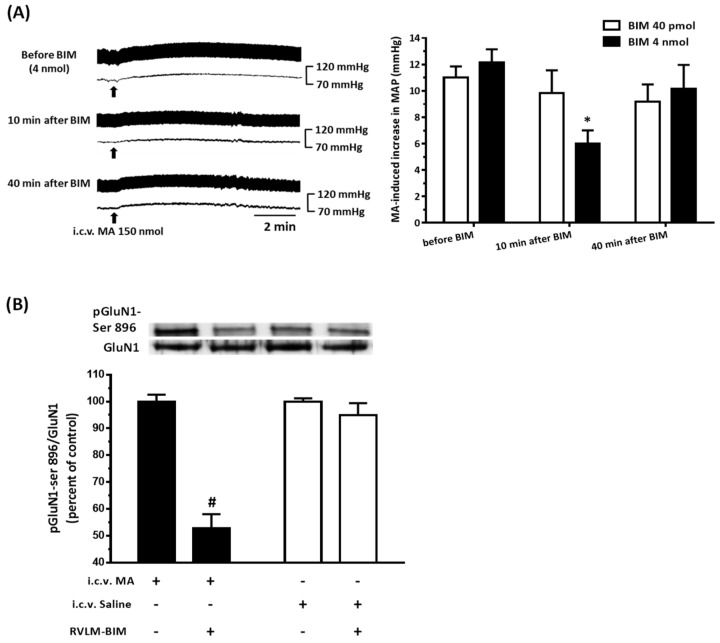
Effects of a selective PKC inhibitor, bisindolylmaleimide (BIM), on the intracerebroventricular (i.c.v.) injection of methamphetamine (MA) induced pressor effects and phosphorylation of GluN1 serine 896 residue in urethane-anesthetized Sprague Dawley (SD) rats. (**A**) Left: Representative recordings show an increase in pulsatile arterial pressure and mean arterial pressure (MAP) following i.c.v. administration of MA (150 nmol) 20 min before and 10 and 40 min after unilateral microinjection of BIM (4 nmol) into the rostral ventrolateral medulla (RVLM) in a SD rat. Right: Bar graph shows an MA-induced increase in MAP before and 10 and 40 min after unilateral microinjection of BIM (40 pmol and 4 nmol) into the RVLM. Values represent the mean + SEM, *n* = 6 animals in each group. * *p* < 0.01 versus corresponding before BIM in one way ANOVA repeated measures followed by Bonferroni’s multiple comparison post-test. (**B**) Top panel shows representative Western blot analysis of the levels of phosphoserine 896 on the GluN1 subunit (pGluN1-Ser 896) and GluN1subunit (GluN1) in the RVLM 5 min after i.c.v. injection of saline or MA 150 nmol. BIM was microinjected into the RVLM unilaterally 10 min before i.c.v. administration of saline or MA. The percentage changes in the ratio of pGluN1-Ser 896 to GluN1 are shown in the bottom graph. The RVLM tissue with a BIM injection side shows RVLM-BIM+ and without a BIM injection side shows RVLM-BIM−. The ratio treatment with saline or MA and without BIM (RVLM-BIM−) are taken as a control of 100%. Values represent the mean + SEM, MA group (*n* = 6), saline group (*n* =3 ). # *p* < 0.0001 versus corresponding control analyzed by unpaired *t*-test.

**Figure 7 biomolecules-10-00149-f007:**
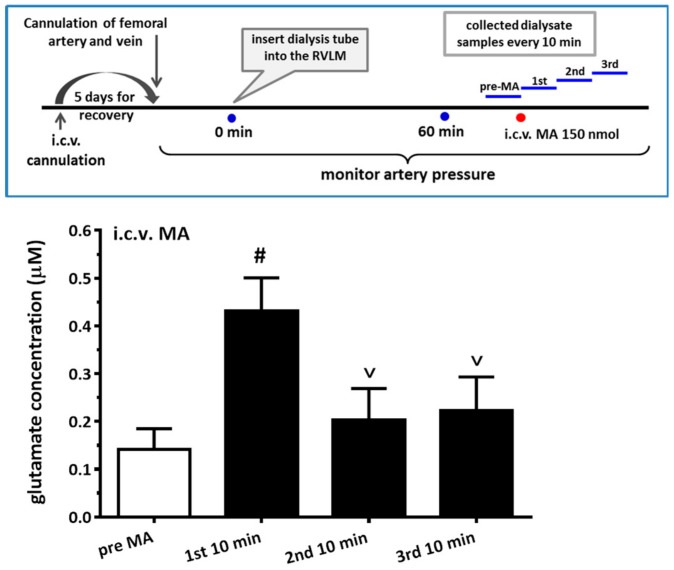
Effects of intracerebroventricular (i.c.v.) injected methamphetamine (MA) on glutamate concentration in the rostral ventrolateral medulla (RVLM) in anesthetized rats. Top panel shows the experimental design for the set of experiments. A dialysis probe was inserted into the RVLM more than 60 min before the first collection of dialysate. At the end, the first collection of dialysate from the RVLM (pre-MA), MA 150 nmol was applied cerebroventricularly through the i.c.v. cannulated tube. The dialysate from the RVLM was collected in 10-min periods after MA was injected (1st, 2nd, and 3rd). Bar graph shows the glutamate concentration in the RVLM dialysate 10-min period before and every 10-min period after i.c.v. injection of 150 nmol MA. Values represent the mean + SEM, *n* = 5 animals, in the set of tests. # *p* < 0.0001 versus pre-MA and ^v^
*p* < 0.0001 versus 1st 10 min analyzed by one way ANOVA repeated measures followed by Bonferroni’s multiple comparison post-test.

**Figure 8 biomolecules-10-00149-f008:**
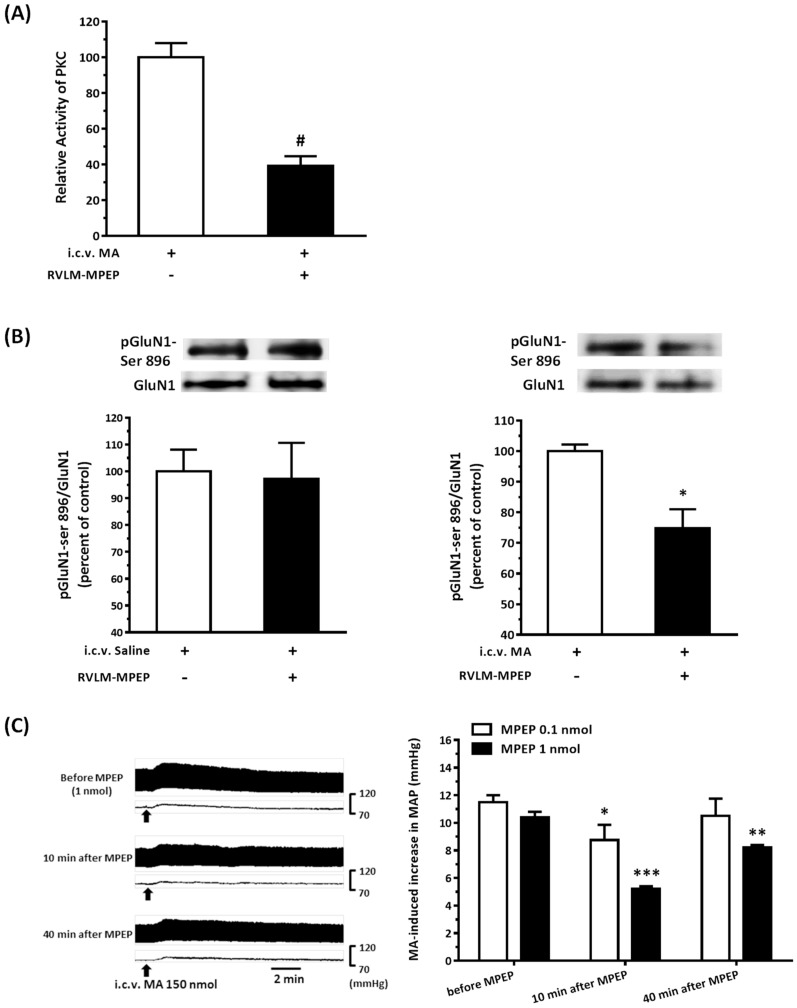
Effects of a selective metabotropic glutamate receptor (mGluR5) antagonist 2-methyl-6-phenylethyl-pyrididine (MPEP) on the intracerebroventricular (i.c.v.) injection of methamphetamine (MA) induced an increase in the PKC activity, phosphorylation of GluN1 serine 896 residue, and pressor effects in urethane-anesthetized Sprague Dawley (SD) rats. (**A**) Bar graph shows the relative activity of PKC in the rostral ventrolateral medulla (RVLM) 5 min after i.c.v. injection of MA 150 nmol. MPEP (1 nmol) was microinjected into the RVLM unilaterally 10 min before i.c.v. administration of MA. The RVLM tissue with the MPEP injection side shows RVLM-MPEP+, and without the MPEP injection side shows RVLM-MPEP−. The relative PKC activity with MA and without MPEP (RVLM-MPEP−) are taken as a control of 100%. Values represent the mean + SEM, *n* = 3 animals in the set of tests. # *p* = 0.0048 versus control analyzed by unpaired *t*-test. (**B**) The top panels show representative Western blot analysis of the levels of phosphoserine 896 on the GluN1 subunit (pGluN1-Ser 896) and the GluN1subunit (GluN1) in the RVLM 5 min after i.c.v. injection of saline (left) or MA (150 nmol) (right) with or without microinjection of MPEP (1 nmol) into the RVLM. The percentage changes in the ratio of pGluN1-Ser 896 to GluN1 are shown in the bottom graphs. The ratio treatment with saline (left) or MA (right) and without MPEP (RVLM-MPEP−) are taken as a control of 100%. Values represent the mean + SEM, *n* = 3 animals in each set of tests. * *p* = 0.0183 versus corresponding control analyzed by unpaired *t*-test. (**C**) Left: Representative recordings show an increase in pulsatile arterial pressure and mean arterial pressure (MAP) following i.c.v. administration of MA (150 nmol) 20 min before and 10 and 40 min after unilateral microinjection of MPEP (1 nmol) into the RVLM in a SD rat. Right: Bar graph shows an MA-induced increase in MAP before and 10 and 40 min after unilateral microinjection of MPEP (0.1 nmol, 1 nmol) into the RVLM. Values represent the mean + SEM, *n* = 4 animals in the MPEP 0.1 nmol group and *n* = 5 animals in the MPEP 1 nmol group. * *p* < 0.01, ** *p* < 0.001, and *** *p* < 0.0001 versus corresponding before MPEP in one way ANOVA repeated measures followed by Bonferroni’s multiple comparison post-test.

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
