# Peer review of "Activation of mGluR5 and NMDA Receptor Pathways in the Rostral Ventrolateral Medulla as a Central Mechanism for Methamphetamine-Induced Pressor Effect in Rats"

_biomolecules, 2020, doi:10.3390/biom10010149_

Round 1

Reviewer 1 Report

A solid study which explores the cellular and molecular mechanism underling methamphetamine (MP)-induced pressor response in either freely moving or urethane anesthetized adult male Sprague Dawley rats.  A multidisciplinary approach including behavioral, microdialysis, pharmacological, and Western blotting were used to bear on the question of signaling mechanism involved in MP induced pressor response.

Experimental procedures were adequately described, results were clearly presented and data were analyzed statistically. MP abuse is on the rise in Taiwan. Viewed in this context, this is a timely and potentially important medico-socio study. Information derived from this study is expected to provide new insight relative to one of the major side effects of MP. 

My decision is accept after minor revisions, mostly becauase of the language issues.

Author Response

A solid study which explores the cellular and molecular mechanism underling methamphetamine (MP)-induced pressor response in either freely moving or urethane anesthetized adult male Sprague Dawley rats. A multidisciplinary approach including behavioral, microdialysis, pharmacological, and Western blotting were used to bear on the question of signaling mechanism involved in MP
induced pressor response.

Experimental procedures were adequately described, results were clearly presented and data were analyzed statistically. MP abuse is on the rise in Taiwan. Viewed in this context, this is a timely and potentially important medico-socio study. Information derived from this study is expected to provide new insight relative to one of the major side effects of MP.
My decision is accept after minor revisions, mostly becauase of the language issues.

Response: In response to the reviewer’s suggestion, we would like to choose the MDPI English Editing Service to edit the manuscript

Reviewer 2 Report

The manuscript by Lai et al. reports on the downstream effects of methamphetamine in the RVLM to produce increases in sympathetic outflow.  Previous reports have shown that methamphetamine increases central sympathetic outflow to increase blood pressure and the current study shows that this results from action at glutamate, in particular mGluR5, receptors in the RVLM.  While this finding isn’t particularly surprising, it does provide additional information regarding the mechanism of action for methamphetamine’s cardiovascular effects.  I do have a few minor concerns that should be addressed.

The author’s state on page 1 that “MA substitutes for noradrenaline binding”.  This may be misleading.  MA acts as a substrate for the norepinephrine (NE) transporter and induces release of NE.

I was confused about the methods and how rats were used in each study.  Were the telemetry rats used for only a single injection of either saline or methamphetamine?  It appears that the same rats were used in both the i.c.v. experiments and the microinjection into the RVLM.  Were some of the rats in these previous studies then sacrificed for the neurochemical studies?  The author’s need to clarify how each rat was treated.

Was there any adaptation to handling and injection procedures for the in vivo studies?

The repeated dose study for i.c.v. injections is important as it apparently shows no acute tolerance.  Why isn’t that data presented?

The author’s state that the lack of effect for direct RVLM injections “implied” that the RVLM is not the direct target for MA.  I think it is more than implied.  It clearly indicates that MA is acting elsewhere and the RVLM effects are downstream of the direct target.

Author Response

The manuscript by Lai et al. reports on the downstream effects of methamphetamine in the RVLM to produce increases in sympathetic outflow.  Previous reports have shown that methamphetamine increases central sympathetic outflow to increase blood pressure and the current study shows that this results from action at glutamate, in particular mGluR5, receptors in the RVLM.  While this finding isn’t particularly surprising, it does provide additional information regarding the mechanism of action for methamphetamine’s cardiovascular effects.  I do have a few minor concerns that should be addressed.

The author’s state on page 1 that “MA substitutes for noradrenaline binding”.  This may be misleading.  MA acts as a substrate for the norepinephrine (NE) transporter and induces release of NE.

Response 1: In response to the reviewer’s suggestion, we have rewritten the sentence as follow: Because of structural similarity, MA acts as a substrate for the noradrenaline transporter and vesicular monoamine transporter 2, leading to….. (page 1, line 38-39)

I was confused about the methods and how rats were used in each study.  Were the telemetry rats used for only a single injection of either saline or methamphetamine? 

Response 2: Yes, the telemetry rats were used for only a single injection of either saline or methamphetamine. We have clarified that in the Materials and Methods section. (page 3, page 98-99)

It appears that the same rats were used in both the i.c.v. experiments and the microinjection into the RVLM. 

Response 3: For detecting the pressor effect of i.c.v. injection of MA, a set of animals was injected with both saline and MA. MA (50, 150 or 500 nmol) was injected 30 minutes after saline injection. For assessing whether the repeated i.c.v. injection of MA did induce reproducible pressor effect, one set of animals was injected with MA (150 nmol) four times at intervals of 30 minutes. The other two sets of animals were used to evaluate the effects of microinjection of kinase inhibitor or mGluR5 antagonist into the RVLM on the pressor effect of i.c.v. injection of MA. We have added the statement in the Materials and Methods section. (page 3, line 126-132)

Were some of the rats in these previous studies then sacrificed for the neurochemical studies? 

Response 4: No, it is not. Each neurochemical study, including Western blotting analysis, ELISA, or HPLC detection, used an independent group of animals. We have clarified that in the Materials and Methods section. (page 2, page 80-82)

The author’s need to clarify how each rat was treated. Was there any adaptation to handling and injection procedures for the in vivo studies?

Response 5: All in vivo experiments were conducted by an expert researcher, Ms. YWW, in our study. The detail handing and injection procedures for the in vivo studies have been added in the Materials and Methods section. (page 3, line 96-101)

The repeated dose study for i.c.v. injections is important as it apparently shows no acute tolerance.  Why isn’t that data presented?

Response 6: In response to the reviewer’s suggestion, we have added a figure for the repeated i.c.v. injection of MA in supplementary data (Figure S1). The figure legend has been added accordingly. We have also written the statement in the TEXT. (page 3, line 128-130; page 6, line 271; page 15, line 581-583 and page 19, line 717-731)

The author’s state that the lack of effect for direct RVLM injections “implied” that the RVLM is not the direct target for MA.  I think it is more than implied.  It clearly indicates that MA is acting elsewhere and the RVLM effects are downstream of the direct target.

Response 7: The authors appreciated the reviewer’s suggestion. We revised the statement as follow: It clearly indicated that MA might act elsewhere in the CNS and the RVLM effects are downstream of the direct target. (page 14, line 538-539)

Response: In response to the reviewer’s suggestion, we would like to choose the MDPI English Editing Service to edit the manuscript.